# RotoGrad: Dynamic Gradient Homogenization for Multitask Learning

## Abstract

GradNorm (Chen et al., 2018) is a broadly used gradient-based approach for training multitask networks, where different tasks share, and thus compete during learning, for the network parameters. GradNorm eases the fitting of all individual tasks by dynamically equalizing the contribution of each task to the overall gradient magnitude. However, it does not prevent the individual tasks' gradients from *conflicting*, i.e., pointing towards opposite directions, and thus resulting in a poor multitask performance. In this work we propose Rotograd, an extension to GradNorm that addresses this problem by dynamically homogenizing not only the gradient magnitudes but also their directions across tasks. For this purpose, Rotograd adds a layer of task-specific rotation matrices that aligns all the task gradients. Importantly, we then analyze Rotograd (and its predecessor) through the lens of game theory, providing theoretical guarantees on the algorithm stability and convergence. Finally, our experiments on several real-world datasets and network architectures show that Rotograd outperforms previous approaches for multitask learning.

## 1 Introduction

While single-task learning is broadly-used and keeps achieving state-of-the-art results in different domains—sometimes beating human performance, there are countless scenarios where we would like to solve several related tasks, we encounter overfitting problems, or the available data is scarce. Multitask learning is a promising field of machine learning aiming to solve—or at least, alleviate—the aforementioned problems that single-task networks suffer from (Caruana, 1993).

Many multitask architectures have emerged throughout the past years (Kokkinos, 2017; Maninis et al., 2019; Dong et al., 2015; He et al., 2017), yet most of them fall under the umbrella of hard parameter sharing (Caruana, 1993). This architecture is characterized by two components: i) a shared backbone which acts as an encoder, and produces an intermediate representation shared across tasks based on the input; and ii) a set of task-specific modules that act as decoders and, using this intermediate representation as input, produce an specialized output for each of the tasks.

These networks have proven to be powerful, efficient, and in many occasions, capable to improve the results of their single-task counterparts. However, they can be difficult to train. Competition among tasks for the shared resources can lead to poor results where the resources are dominated by a subset of tasks. Internally, this can be traced to a sum over task gradients of the form $L = \sum_k \nabla L_k$ with respect to the backbone parameters. In this setting two undesirable scenarios may occur, as depicted in Figure 1a: i) a subset of tasks dominate the overall gradient evaluation due to magnitude differences across task gradients; and ii) individual per-task gradients point towards different directions in the parameter space, cancelling each other out.

This problem can be seen as a particular case of *negative transfer*, a phenomenon describing when "sharing information with unrelated tasks might actually hurt performance" (Ruder, 2017). Under the assumption that the problem comes from task unrelatedness, one solution is to use only related tasks. Several works have explored this direction (Thrun & O'Sullivan, 1996; Zamir et al., 2018), see for example Standley et al. (2019) for a recent in-depth study on task relatedness.

Another way of tackling the problem is by weighting the task dynamically during training, the motivation being that with this approach the task contribution can me manipulated at will and a

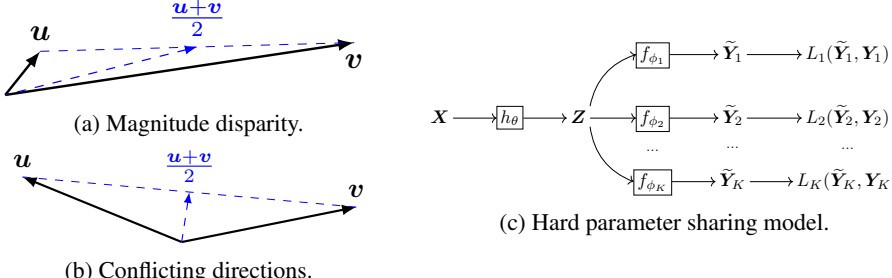

(a) Magnitude disparity.

(b) Conflicting directions.

(c) Hard parameter sharing model.

Figure 1: Examples of: (a) neglecting one vector due to magnitude difference; (b) partial cancellation of two vectors with equal magnitude; (c) a hard parameter sharing architecture.

good balance between gradient magnitudes can be found. Two important works on this direction are GradNorm (Chen et al., 2018), which we will cover in detail later, and Kendall et al. (2018), which adopts a probabilistic approach to the problem. For example Guo et al. (2018), adapts the weights based on their "difficulty" to be solved to prioritize harder tasks.

Furthermore, a series of papers attempting to solve the conflicting gradients problem in different settings have recently emerged. For example, Flennerhag et al. (2019) deals with a similar problem in the context of meta-learning, and Yu et al. proposes two different heuristic solutions for reinforcement learning. In the context of multitask learning, Levi & Ullman (2020) adopts an image-dependent solution that uses a top-down network, and Maninis et al. (2019) applies adversarial training to deal with conflicting gradients, making use of double-backpropagation which poorly scales.

The key contributions of this paper are the following:

- The introduction of Rotograd, an extension of GradNorm that homogenizes both the magnitude and the direction of the task gradients during training. Different experiments support this statement, showing the performance gain provided by Rotograd.

- A novel interpretation of GradNorm (and Rotograd) as a gradient-play Stackelberg game, which allow us to draw theoretical guarantees regarding the stability of the training process as long as GradNorm (Rotograd) is an asymptotically slower learner than the network's optimizer.

- A new implementation of GradNorm where is it applied to the intermediate representation instead of a subset of the shared parameters. This way remove the process of choosing this subset. Besides, we give some guarantees regarding the the norm of the gradient with respect to *all the shared parameters*, and empirically show that results are comparably good to the usual way of applying GradNorm.

## 2 BACKGROUND

### 2.1 MULTITASK LEARNING

Let us consider $K$ different tasks that we want to simultaneously learn using a gradient-based approach. All of them share the same input dataset $\boldsymbol{X} \in \mathbb{R}^{N \times D}$, where each row $\boldsymbol{x}_n \in \mathbb{R}^D$ is a $D$-dimensional sample. Additionally, each task has its own outputs $\boldsymbol{Y}_k \in \mathbb{R}^{N \times I_k}$, and loss function $\ell^k : \mathbb{Y}_k \times \mathbb{Y}_k \to \mathbb{R}$.

As mentioned in Section 1 and illustrated in Figure 1c, the model is composed of a shared backbone and task-specific modules. Upon invoking the model, first the input $\boldsymbol{X} \in \mathbb{R}^{B \times D}$ is passed through the *backbone*, $h_\theta : \mathbb{X} \to \mathbb{Z}$, which produces a common intermediate representation of the input, $\boldsymbol{Z} \in \mathbb{R}^{B \times d}$, where $B$ is the batch size and $d$ the size of the *shared intermediate space*. Then, each task-specific module, $f_{\phi_k} : \mathbb{Z} \to \mathbb{Y}_k$, using the intermediate representation $\boldsymbol{Z}$ as input, predicts the desired outcome for its designated task, $\widehat{\boldsymbol{Y}}_k$.

As the focus of this work concerns only the shared parameters, we can simplify the notation regarding the task-specific pipeline—that is, the branching paths shown in Figure 1c—by writing the single-point $k$-th loss evaluation at time $t$ as $\ell_i^k(t)$, instead of $\ell^k(f_{\phi_k^t}(\boldsymbol{z}_i^t))$, and denoting by $L_k(t) := \sum_i \ell_i^k(t)$ the total loss for the $k$-th task.

When it comes to training, it is necessary to use a single loss function in order to backpropagate through it. In our setting, however, there is a loss function $\ell_k$ per task. A common approach is to combine these losses by taking a weighted sum, so that the training objective is to minimize the loss function $L(t) := \sum_k \omega_k L_k(t)$. Therefore, the common parameters $\theta$ are updated following the direction of the gradient

$$\nabla_\theta L(t) = \sum_{k=1}^K \omega_k \nabla_\theta L_k(t). \tag{1}$$

As we noted in Section 1, Equation 1 turns out to be the main source of problems since *the loss functions—and thus the tasks—have been artificially coupled*. The magnitude of these task-specific gradients represent how much they contribute to the parameter update: if some of them are relatively small (large), their tasks will be overlooked (prioritized) during the training procedure, as Figure 1a exemplifies. More subtly if gradients from different tasks point to completely different regions of the parameter space, they may cancel each other out, resulting in a direction update potentially harmful for these tasks, as shown in Figure 1b.

## 2.2 STACKELBERG GAMES

Let us now introduce some game theoretical concepts necessary for the following sections. A Stackelberg game (Fiez et al., 2019) is a type of *asymmetric game* where two players play alternately. The first player is the follower $\mathcal{F}$, whose sole objective is to minimize its own loss function. The other player is the leader $\mathcal{L}$ who also attempts to minimize its own loss function, but having access to additional information regarding which will be the best response the follower could employ in the next turn. In mathematical terms,

$$
\begin{aligned}
\mathcal{L}\text{eader: } &\min_{x_l \in X_l} \{\mathcal{L}(x_l, x_f) |\ x_f \in \operatorname*{argmin}_{y \in X_f} \mathcal{F}(x_l, y)\}, \\
\mathcal{F}\text{ollower: } &\min_{x_f \in X_f} \mathcal{F}(x_l, x_f),
\end{aligned}
\tag{2}
$$

where $x_l$ and $x_f$ are the actions of the leader and the follower, respectively.

Another important concept in game theory is that of a local equilibrium point. In layman's terms, a local equilibrium point in a game between two players is one in which both players have reached a state that satisfies them, meaning that there is no available move improving any of the player scores and no further actions are taken. In our case we are interested in the following definition:

**Definition 2.1** (differential Stackelberg equilibrium). The pair of points $x_l^* \in X_l$, $x_f^* \in X_f$, where $x_f^* = r(x_l^*)$ is implicitly defined by $\nabla_{x_f} \mathcal{F}(x_l^*, x_f^*) = 0$, is a differential Stackelberg equilibrium if $\nabla_{x_l} \mathcal{L}(x_l^*, r(x_l^*)) = 0$, and $\nabla_{x_l}^f \mathcal{L}(x_l^*, r(x_l^*))$ is positive definite.

As a final remark, note that in this particular concept of equilibrium both players reach a critical point (local optimum). Even more, when both players optimize their objectives using a gradient-based approach and the leader is a slower learner—meaning having an asymptotically smaller learning rate, both players reach a local minimum on the equilibrium point (Fiez et al., 2019).

## 3 THE GAME OF GRADNORM

GradNorm (Chen et al., 2018) was conceived with a simple idea in mind: equalizing the contribution of each task to during training such that they all are learned at the same rate. This idea, as observed in Equation 1, means making the gradient magnitudes relatively equal across all tasks. GradNorm achieves this by considering the task weights $\omega_k$ as parameters that are modified during training.

These parameters are trained alongside the original network parameters via gradient descent, yet their gradients are obtained by minimizing a specific loss function:

$$L_{\text{grad}}^k(\omega_k, \theta) := \left| \omega_k ||\nabla_W L_k(t)||_2 - \frac{1}{K} ||\nabla_W L(t)||_2 \times [r_k(t)]^\alpha \right|, \tag{3}$$

where $W$ is a subset of the shared network parameters $\theta$, the weights are corrected such that $\sum_k \omega_k = K$ always, $\alpha$ is a hyperparameter and $[r_k(t)]^\alpha$ balances the learning speed per task, and the average gradient magnitude (that is, the target magnitude) is treated as a constant.

Using the concepts from Section 2.2, we can reinterpret GradNorm as a Stackelberg game. Namely, in this new point of view the follower would be the player whose objective is to optimize the parameters of the backbone, whereas GradNorm would play the role of the leader, optimizing the task weights in order to equalize the gradient magnitudes:

$$\mathcal{L}\text{eader: } \min_{\{\omega_k\}_k} \sum_k L_{\text{grad}}^k(\omega_k, \theta), \quad \mathcal{F}\text{ollower: } \min_\theta \sum_k \omega_k L_k(\theta). \tag{4}$$

Note that there is a fundamental difference between Equation 2 and 4: the extra information available to the leader is not explicitly written in its optimization problem. This is due to the more subtle nature of this bit of information. *The actual objective of the leader is not to equalize the magnitudes at the current step $t$, but at $t + 1$, since that is when the effect of updating $\omega_k$ will take effect.* Therefore, the extra information that the leader possesses is that the follower updates its parameters in a smooth way and the gradient magnitude between consecutive steps are similar.

This is, to the best of our knowledge, the first time GradNorm has been interpreted as a (stochastic) gradient-based Stackelberg game. The main advantage of this point of view is that, based on the results presented by Fiez et al. (2019), we can provide theoretical guarantees regarding the convergence of the learning dynamics between the leader (GradNorm) and the follower (network's optimizer). Specifically, adapted from Fiez et al. (2019):

**Proposition 3.1.** *Suppose a finite-time (stochastic) gradient-based Stackelberg where the learning rate of the leader tends to zero on a faster timescale than the follower, that is, $\alpha_l(t) = o(\alpha_f(t))$. Then, the game will with high probability converge to a neighborhood of a stable differential Stackelberg equilibrium.*

This previous statement means—in terms of the training dynamics of the network—that, as long as the leader is the slow learner, the parameters of the leader will stabilize at some point and the network parameters will end up in local optimum. *In other words, the training will be stable.*

## 4 ROTOGRAD

### 4.1 $\mathbf{Z}$ AS THE CRITICAL SPACE

As an additional remark to Section 3, we believe that it is worth noting that GradNorm is applied to only a subset of the network parameters $W \subset \theta$ for efficiency reasons. The selection of such subset is left to the practitioner and, to the best of our knowledge, there exists no intuition regarding which parts of the backbone are preferred to be selected as $W$. In this work, we propose to apply GradNorm on the last shared representation of the network, which as shown in Figure 1c, we refer to as $\mathbf{Z}$. In other words, we assume that $W = \mathbf{Z}$.

Since the size of $\mathbf{Z}$ tends to be orders of magnitude smaller than the total number of shared parameters $\theta$, that is, $B \times d << |\theta|$, this choice saves on computation. Moreover, we argue that this selection is the most logical one since the intermediate representation $\mathbf{Z}$ is the last shared representation across all tasks, as shown in Figure 1c. In order to further motivate this selection of $W$ we provide and proof a proposition in Appendix A.1 stating that, under the assumption that $\nabla_\theta \mathbf{Z}$ has linearly independent columns (which is mild since $\theta >> d$), the magnitude of the gradient with respect to the (arbitrarily many) shared parameters $\theta$ is bounded on both sides, which is in stark contrast with the lack of guarantees when $W$ is selected as a (strict) subset of $\theta$.

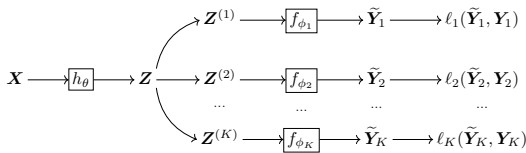

Figure 2: Extended hard parameter sharing model with task-specific representations.

## 4.2 ROTOGRAD: A BETTER LEADER

In Section 3 we saw that GradNorm's objective is to equalize the gradient magnitude so that, in the next step, the shared parameters are updated according to the necessities of all the tasks. GradNorm does not address the conflicting gradients problem (Figure 1b) in the total gradient computation (Equation 1), which can be as detrimental as the one of magnitude disparity (Figure 1a). Fortunately, the reinterpretation of GradNorm as a Stackelberg game presented in Section 2.2 allow us to easily fix this problem. More specifically, *we introduce a new leader that aims to homogenize both the gradient magnitudes and directions for all tasks. We refer to the resulting algorithm as Rotograd.*

Rotograd extends GradNorm by additionally expanding the model's architecture as shown in Figure 2. As before, $\boldsymbol{Z} \in \mathbb{R}^{B \times d}$ represents the shared intermediate representation, and now $\boldsymbol{Z}^k \in \mathbb{R}^{B \times d}$ denotes task-specific intermediate representations of $\boldsymbol{Z}$, one per task. This new space $\mathbb{Z}^k$ is a shifted-and-rotated version of $\mathbb{Z}$, such that the $i$-th instance of $\boldsymbol{Z}$ is transformed as $\boldsymbol{z}_i^k = R_k \boldsymbol{z}_i + \boldsymbol{d}_k$, where $R_k \in SO(d)$ is a rotation matrix and $\boldsymbol{d}_k \in \mathbb{R}^d$ a translation vector.

For the $k$-th task at step $t$, Rotograd optimizes the weights $\omega_k$ as GradNorm, and finds the rigid motion that minimizes the distance between the *the next set of shared points* $\{\boldsymbol{z}_i^{t+1}\}_{i=1}^B$ *to the next set of points we would have reached if only that task were being trained* $\{\boldsymbol{z}_i^{k,t+1}\}_{i=1}^B$. Note that the additional knowledge of the leader plays a role, as it knows the next evaluation points given the current ones, concept that we represent using two oracle functions[1] $\mathcal{O}(\boldsymbol{z}_i^t)$ and $\mathcal{O}_k(\boldsymbol{z}_i^{k,t})$ for the multi-task and single-task next points, respectively.

This new objective function can be mathematically expressed as

$$L_{\text{rot}}^k(R_k, \boldsymbol{d}_k, \theta) := \frac{1}{B} \sum_{i=1}^B ||R_k \mathcal{O}(\boldsymbol{z}_i^t) + \boldsymbol{d}_k - \mathcal{O}_k(\boldsymbol{z}_i^{k,t})||_2^2, \tag{5}$$

resulting in the Stackelberg game:

$$\begin{aligned}
&\mathcal{L}\text{eader:} \underset{\{\omega_k, R_k, \boldsymbol{d}_k\}_k}{\text{minimize}} \sum_k \left[ L_{\text{rot}}^k(R_k, \boldsymbol{d}_k, \theta) + L_{\text{grad}}^k(\omega_k, \theta) \right], \\
&\mathcal{F}\text{ollower:} \underset{\theta}{\text{minimize}} \sum_k \omega_k L_k(\theta).
\end{aligned} \tag{6}$$

## 4.3 SOLVING THE NEW OBJECTIVE FUNCTION

At first glance solving the newly-formulated problem $L_{\text{rot}}$ seems quite challenging: we cannot write down the oracle functions, and it is a constrained optimization problem since $\mathbb{R}_k$ is an orthogonal matrix. In this section we show how to approximately solve this problem in closed form.

Recall that, at the current step $t$, we can obtain any current intermediate point in the shared space as $\boldsymbol{z}_i^t = h_{\theta^t}(\boldsymbol{x}_i^t)$, and its task-specific equivalent as $\boldsymbol{z}_i^{k,t} = R_k^t \boldsymbol{z}_i^t + \boldsymbol{d}_k^t$. To simplify notation, let us denote the gradient of the $k$-th task with respect to $\boldsymbol{z}^k$ at step $t$ by $g_i^{k,t} := \nabla_{\boldsymbol{z}^k} \ell_k(\boldsymbol{z}_i^{k,t})$, and the gradient of the weighted loss with respect to $\boldsymbol{z}$ at step $t$ by $g_i^t := \nabla_{\boldsymbol{z}} \ell(\boldsymbol{z}_i) = \sum_k \omega_k R_k^\top g_i^{k,t}$.

As mentioned before, we cannot in general write the oracle functions in closed form. However, since we assume that the network parameters are optimized using a first-order optimization method, we can *approximate* the oracle functions by assuming that the next point will follow the gradient

---

[1]These could be simulated by training the model with all and a single task at each step, respectively.

direction, that is, we can approximate the oracles as

$$\mathcal{O}(\boldsymbol{z}_i^t) \approx \boldsymbol{z}_i^t - \beta g_i^t, \qquad \text{and} \qquad \mathcal{O}_k(\boldsymbol{z}_i^{k,t}) \approx \boldsymbol{z}_i^{k,t} - \beta g_i^{k,t}, \qquad (7)$$

where we treat the step-size $\beta$ as an unknown variable.

By plugging this approximation to the oracles into the loss function in Equation 5 and using the law of cosines we can approximately solve $\min_{R_k, \boldsymbol{d}_k} L_{\text{rot}}^k(R_k, \boldsymbol{d}_k, \theta)$ by solving instead

$$\min_{R_k \in O(d)} \frac{1}{B} \sum_{i=1}^{B} ||R_k g_i^t - g_i^{k,t}||_2^2 \quad \text{and} \quad \min_{\boldsymbol{d}_k \in \mathbb{R}^d} \frac{1}{B} \sum_{i=1}^{B} ||R_k \boldsymbol{z}_i^t + \boldsymbol{d}_k - \boldsymbol{z}_i^{k,t}||_2^2, \qquad (8)$$

which are two least-square problems with closed form solutions $R_k^*$ and $d_k^*$. Details in Appendix A.2.

Note that the first problem in Equation 8 looks for the orthogonal matrix that *minimizes the gradient conflict between the weighted gradients $g_i^t$ and the gradients of the $k$-th task $g_i^{k,t}$*, which was our main objective for this extension. The second problem looks for the best way of translating the current points in the common space $\boldsymbol{z}_i^t$ to the current points that were plugged into the task-specific module for the $k$-th task $\boldsymbol{z}_i^{k,t}$.

Similarly to what we did in Section 3 we can apply again Proposition 3.1, thus *providing theoretical guarantees on the stable behaviour of the algorithm as long as Rotograd has a stronger learning rate decay than the network's optimizer*. In order for this to hold, we should adopt a gradient-descent approach and, since we have an approximation to the solution of the problem, we set the gradient of our parameters to the difference between our solution and its current value. For example, in the case of applying SGD to the parameter $R_k$, the update rule would be of the form $R_k^{t+1} = R_k^t - \alpha(R_k^t - R_k^*)$ where, afterwards, we should make sure that $R_k^{t+1}$ is a rotation matrix.

An overview of the proposed Rotograd algorithm can be found in Algorithm 1.

---

**Algorithm 1** Forward and update methods for the $k$-task using Rotograd.

---

1: **procedure** FORWARD_K($\boldsymbol{z}_i^t$)
2:     **return** $R^k \boldsymbol{z}_i^t + \boldsymbol{d}_k$                        ▷ Transform to task-specific space $\mathbb{Z}^k$
3: **end procedure**
4:
5: **procedure** UPDATE_LOSS_ROT_K($\{\boldsymbol{z}_i^t, g_i^t, \boldsymbol{z}_i^{k,t}, g_i^{k,t}\}_i$)
6:     $R_k^* \leftarrow \operatorname{argmin}_{R_k} \frac{1}{B} \sum_{i=1}^{B} ||R_k g_i^t - g_i^{k,t}||_2^2$      ▷ Solve conflicting gradient problem
7:     $\boldsymbol{d}_k^* \leftarrow \operatorname{argmin}_{\boldsymbol{d}_k} \frac{1}{B} \sum_{i=1}^{B} ||R_k \boldsymbol{z}_i^t + \boldsymbol{d}_k - \boldsymbol{z}_i^{k,t}||_2^2$      ▷ Solve translation problem
8:
9:     $\boldsymbol{d}_k^{t+1} \leftarrow \boldsymbol{d}_k^t - \alpha(\boldsymbol{d}_k^t - \boldsymbol{d}_k^*)$               ▷ Update rule for $\boldsymbol{d}_k$
10:     $R_k^{t+1} \leftarrow R_k^t - \alpha(R_k^t - R_k^*)$                  ▷ Update rule for $R_k$
11:
12:     $U, S, V^\top \leftarrow \text{SVD}(R_k^{t+1})$
13:     $R_k^{t+1} \leftarrow U V^\top$                       ▷ Ensure that $R_k^{t+1}$ is a rotation
14: **end procedure**

---

## 5 EXPERIMENTS

### 5.1 QUALITATIVE RESULTS

In order to get a better understanding of Rotograd, we provide two simple synthetic examples that show both, its effect on the shared intermediate space $\boldsymbol{Z}$, and its effect on the final predictions.

First, we consider two different settings where we use a two-layer feed-forward NN with 10 hidden units to solve two regression tasks. The first setting problem, illustrated in Figure 3a, shows an example where not correcting for conflicting gradients may completely prevent from learning (i.e., none of the tasks is optimized), while Rotograd converges to the global optima of both tasks. The second setting in Figure 3b shows that, due to differences in the gradient magnitudes, only one of

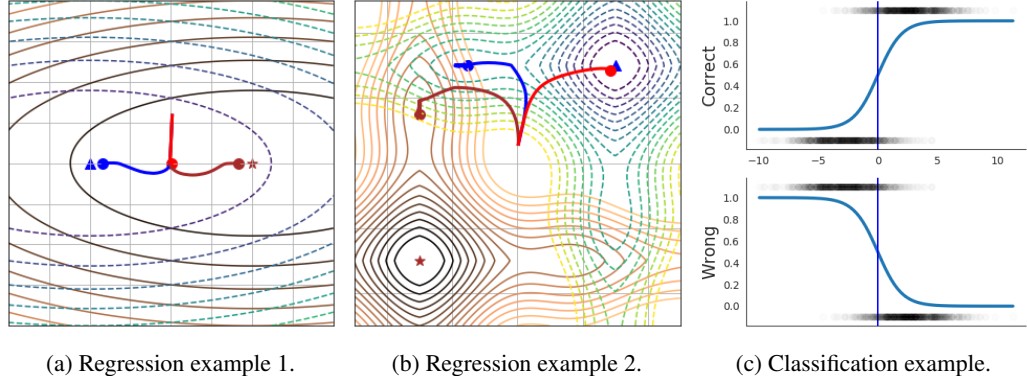

| (a) Regression example 1. | (b) Regression example 2. | (c) Classification example. |

Figure 3: Figures (a) and (b) show the trajectories followed when using Rotograd in the spaces $Z^1$ (blue) and $Z^2$ (brown), versus not using it (red). Circular markers mean the final point of the trajectory, whereas the triangle and star mark the global optimum of each task. Figure (c) shows the deep logistic regression classifier learned using Rotograd to predict a binary label and its (Correct) and its negated (Wrong).

the tasks is learned when Rotograd is not applied. It is worth-mentioning as well that Rotograd is still subject to the nature of stochastic optimization, and thus being subject to fall into local optima, as shown in Figure 3b.

As an extreme example, we consider two classification tasks, which aim to predict opposite labels. As before, we use a feed-forward backbone without task-specific networks, and a logistic output for each task. In this case, the two tasks, and therefore their gradients with respect to the network parameter, are in conflict. Thus, without additional model flexibility (i.e., without our rotation layer) the network cannot learn any of the tasks. In contrast, Rotograd is able to rotate the space and learn that one function is the opposite of the other, learning an accurate classifier, as shown in Figure 3c.

## 5.2 QUANTITATIVE RESULTS

**Dataset and tasks.** Similar to Sener & Koltun (2018), we employ as dataset a multitask version of MNIST (LeCun et al., 1998) that has been expanded to contain instead two digits, one in each side of the picture. Regarding the tasks, we learn five different tasks: i) classify the left-side digit; ii) classify the right-side digit; iii) predict the sum of the two digits; iv) predict their product; and v) predict the number of pixels that are activated in the picture. The idea behind these tasks is no other than having a set of tasks that are correlated, highly correlated, and uncorrelated.

**Architecture.** We use as backbone a modified version of LeNet (LeCun et al., 1998) where the number of parameters has been greatly reduced in order to not have enough parameters to learn all the tasks perfectly, and thus, try to force them to collaborate in order to improve their results. The size of $Z$ is $d = 25$. For the task-specific parts we make use of a single layer per task, followed by a sigmoid activation function in the case of solving a classification problem. Regarding loss functions, we use mean-squared error for the regression tasks and negative log-likelihood for the classification ones. Furthermore, we divide all losses by their loss at the initial step to normalize them.

**Training.** We train all models for $150$ epochs using a batch size of $1024$ and Adam (Kingma & Ba, 2014) as optimizer. All experiments are averaged over ten different runs. The learning rate of the network parameters are is set to $0.001$ and the rest of hyperparameters are selected via grid search.

**Metrics.** We employ the mean-squared error for regression tasks and accuracy for classification tasks. Besides, as in Vandenhende et al. (2019), we include an additional metric that summarizes the overall performance of the multitask network:

$$\Delta_m = \frac{1}{K} \sum_{k=1}^{K} (-1)^{l_k} \frac{M_{m,k} - M_{s,k}}{M_{s,k}}, \tag{9}$$

| Exp. decay | Left ↑ | Right ↑ | Sum ↓ | Multiply ↓ | Density ↓ | $\Delta_m$ ↑ |
|---|---|---|---|---|---|---|
| 0.9 | $89.46 \pm 00.57$ | $85.90 \pm 00.65$ | $4.64 \pm 0.25$ | $133.97 \pm 7.20$ | $0.26 \pm 0.06$ | $0.18 \pm 0.06$ |
| 0.99 | $88.99 \pm 00.81$ | $84.92 \pm 00.97$ | $4.58 \pm 0.16$ | $135.19 \pm 4.35$ | $0.24 \pm 0.05$ | $0.18 \pm 0.06$ |
| 0.999 | $85.51 \pm 01.55$ | $80.09 \pm 01.86$ | $4.79 \pm 0.18$ | $142.07 \pm 3.86$ | $0.24 \pm 0.08$ | $0.15 \pm 0.06$ |
| 0.9999 | $84.56 \pm 01.65$ | $79.46 \pm 01.80$ | $4.81 \pm 0.23$ | $142.64 \pm 4.07$ | $0.24 \pm 0.06$ | $0.15 \pm 0.06$ |
| 1.0 | $83.55 \pm 01.69$ | $79.31 \pm 02.43$ | $4.88 \pm 0.20$ | $143.41 \pm 3.17$ | $0.25 \pm 0.13$ | $0.13 \pm 0.09$ |
| $R_k^{t+1} = R_k^*$ | $64.73 \pm 04.83$ | $58.55 \pm 05.94$ | $6.11 \pm 0.26$ | $193.93 \pm 9.53$ | $0.89 \pm 0.21$ | $-0.23 \pm 0.21$ |

Table 1: Task performance on MNIST for different leader's learning speeds. Note that multitask performance gradually worsens as the leader learns faster, being the case where no iterative updates are performed clearly unstable.

| LeNet | Method | Left ↑ | Right ↑ | Sum ↓ | Multiply ↓ | Density ↓ | $\Delta_m$ ↑ |
|---|---|---|---|---|---|---|---|
| original | *uniform* | $96.40 \pm 00.17$ | $94.99 \pm 00.16$ | $1.74 \pm 0.39$ | $38.07 \pm 1.21$ | $0.28 \pm 0.07$ | $0.13 \pm 0.20$ |
| | *rotograd* | $96.45 \pm 00.14$ | $94.89 \pm 00.16$ | $1.67 \pm 0.33$ | $38.70 \pm 1.89$ | $0.27 \pm 0.08$ | $0.15 \pm 0.18$ |
| reduced | *uniform* | $90.15 \pm 00.53$ | $86.65 \pm 00.41$ | $5.14 \pm 0.33$ | $149.21 \pm 5.97$ | $0.51 \pm 0.02$ | $0.06 \pm 0.11$ |
| | *rotograd* | $89.01 \pm 00.87$ | $84.62 \pm 01.19$ | $4.54 \pm 0.19$ | $134.95 \pm 5.92$ | $0.23 \pm 0.04$ | $0.18 \pm 0.06$ |

Table 2: Task performance on MNIST for Rotograd and a baseline method using the same architecture with different number of parameters. As observed, the effect of negative transfer diminishes as the model capacity increases.

where $M_{m,k}$ and $M_{s,k}$ are the performance on the $k$-task for the multitask and single task networks, respectively, and $l_k$ is 1 if a lower value means better performance and 0 otherwise. This metric simply represents the average performance gained per task.

### 5.2.1 ABLATION STUDY

In order to better understand different aspects of the proposed algorithm, we perform a simple ablation study where only the quantities of interest are modified.

**Slow-learner leader.** To understand the negative effect that having a fast-learner leader can have in the results, we set the initial learning rate of the leader to 0.02 and, at each iteration, exponential decay is applied to it, being the decay rate the quantity that changes across experiments. Moreover, and as an extreme case, we consider the case where the parameters of Rotograd are directly set to their approximated closed-form solutions[2] ($R_k^{t+1} = R_k^*$), which would further justify the use of iterative updates as introduced in Section 4.3.

Table 1 show the task performance for different types of learning speeds. Results show that there is a clear trend relating the multitask performance and the decay rate applied to the leader. As this value gets closer and closer to one, the leader becomes effectively a faster learner than the follower, and instabilities in the training process (which is easily observed in the classification tasks' variance) result in worst performance. As seen in the last row of Table 1 (and in Appendix A.3), when iterative updates are omitted the training becomes extremely unstable.

**Model capacity.** As the number of shared parameters increases, it is sensible to think that the model will have a better time solving the same set of tasks, as it has more capacity to store the same information and shared parameters are more likely not to be shared. Therefore, we expect the effect of negative transfer to be less noticeable on bigger models. To verify this, we run Rotograd as well as a baseline method where uniform weights on two different architectures: the original LeNet (LeCun et al., 1998); and the reduced version mentioned at the beginning of the section. Results shown in Table 2 confirm this intuition since results on the original architecture are statistically identical for both methods, yet in the reduced version negative transfer has more impact and Rotograd is able to alleviate its effect.

---

[2]Note that this is equivalent to using SGD as optimizer and setting the learning rate to one.

| Method | Left ↑ | Right ↑ | Sum ↓ | Multiply ↓ | Density ↓ | $\Delta_m$ ↑ |
|---|---|---|---|---|---|---|
| *single task* | $93.50 \pm 00.47$ | $90.65 \pm 00.46$ | $6.44 \pm 4.63$ | $159.08 \pm 6.16$ | $1.62 \pm 1.78$ | - |
| *uniform* | $90.15 \pm 00.53$ | $86.65 \pm 00.41$ | $5.14 \pm 0.33$ | $149.21 \pm 5.97$ | $0.51 \pm 0.02$ | $0.06 \pm 0.11$ |
| *rotograd* | $89.01 \pm 00.87$ | $84.62 \pm 01.19$ | $4.54 \pm 0.19$ | $134.95 \pm 5.92$ | $0.23 \pm 0.04$ | $0.18 \pm 0.06$ |
| *rotograd-sgd* | $87.30 \pm 00.36$ | $83.48 \pm 00.69$ | $5.08 \pm 0.31$ | $161.86 \pm 8.71$ | $0.30 \pm 0.09$ | $0.12 \pm 0.04$ |
| *gradnorm $Z$* | $89.15 \pm 00.58$ | $85.32 \pm 00.72$ | $5.07 \pm 0.30$ | $144.91 \pm 8.51$ | $0.31 \pm 0.07$ | $0.14 \pm 0.05$ |
| *gradnorm $W$* | $89.60 \pm 00.52$ | $85.72 \pm 00.70$ | $5.13 \pm 0.24$ | $151.58 \pm 8.01$ | $0.44 \pm 0.28$ | $0.13 \pm 0.05$ |
| *d-grad* | $88.71 \pm 00.69$ | $84.96 \pm 00.96$ | $4.93 \pm 0.38$ | $144.54 \pm 9.88$ | $0.60 \pm 0.54$ | $0.08 \pm 0.16$ |
| *adversarial* | $89.37 \pm 00.60$ | $85.58 \pm 00.53$ | $5.60 \pm 0.45$ | $160.00 \pm 6.98$ | $1.65 \pm 1.06$ | $-0.25 \pm 0.43$ |

Table 3: Test results on the MNIST dataset for different methods. The arrow on the each column's title describes whether higher is better or the other opposite.

### 5.2.2 COMPARING DIFFERENT METHODS

We consider two different baselines: *single task*, which learns each of the tasks independently (and which will be also used to compute $\Delta_m$); and *uniform*, which simply assumes uniform weights and there is no leader involved. Besides, we consider two different versions of GradNorm and Rotograd. First, to understand the effect of using $W = Z$ instead of a subset of $\theta$, we include these two methods: *gradnorm $Z$* and *gradnorm $W$*, respectively. Second, to understand the effect of using $R_k^t - R_k^*$ as update direction instead of the usual gradient, we compare the proposed Rotograd with *rotograd-sgd*, a variation that also optimizes Equation 8 but relying on automatic differentiation to compute the update direction. Finally, we compare with two different methods: *adversarial* (Maninis et al., 2019), which uses adversarial training to correct the gradient directions and a two-layer feed-forward neural network as discriminator; and *d-grad*, an adaption of the method proposed by Yu et al. for hard-parameter sharing that simply adds a learnable affine transformation per task before each task-specific module.

Figure 3 summarizes the results obtained. We observe how Rotograd outperforms the rest of methods with regards to the overall improvement performance $\Delta_m$. Rotograd is able to drastically increase the performance on the regression tasks with just a slight trade-off on the classification ones. Not only we can say that there is an improvement, but also that *approximately solving the problem in Equation 13 in closed-form is important* for it, as *rotograd-sgd* is far from Rotograd's results. When it comes to GradNorm, Table 3 shows that there is no disadvantage on applying the algorithm to $Z$, thus removing the selection of the hyperparameter $W$. In Appendix A.3 and A.4 we show further results regarding the metrics and cosine similarity evolution during traning, as well as some preliminary results on a larger dataset such as ChestX-ray14 (Rajpurkar et al., 2017).

Finally, it is worth mentioning that *d-grad*, and *adversarial* perform considerably worse. Our intuition for these results are that: i) *d-grad* adds more parameters to each task-specific modules, but since the per-task capacity is not the problem, it does not result into much of an improvement; and ii) *adversarial* suffers from training instability, as it is common with other adversarial training techniques such as Generative Adversarial Networks (Arjovsky & Bottou, 2017). Additional experiments on the ChestX-ray14 dataset are included in Appendix A.4, where we observed similar patterns in the results for the different methods, being the results of GradNorms and Rotograd comparable, as it seems that such dataset does not suffer from the conflicting gradient problem.

## 6 CONCLUSIONS

In this work, we have proposed a novel learning algorithm for multitask learning, Rotograd, which aims to minimize negative transfer among tasks by homogenizing the gradient magnitudes and directions across tasks. Our starting point here has been GradNorm, which we have reinterpreted as a Stackelberg game where GradNorm act as the leader and the network optimizer as the follower. With this novel interpretation, we have proposed Rotograd, and derived stability and convergence guarantees for both Rotograd and GradNorm.

As future work, we plan to extend Rotograd to related scenarios, such as soft parameter sharing networks and meta-learning, which could gratefully benefit from using our algorithm. We also leave for future research trying Rotograd in heavier experiments that are affected by negative transfer, as it could happen in the field of computer vision.

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

# A   APPENDIX

## A.1   BOUNDING THE MAGNITUDE GRADIENT W.R.T THE SHARED PARAMETERS

**Proposition A.1.** *If for every task* $||\omega_k \nabla_{\boldsymbol{Z}} L_k(t)||_2 = C$*, where* $C$ *is a constant value, and the rank of* $\nabla_\theta \boldsymbol{Z}$ *is* $d$*, then:*

$$\frac{C}{||(\nabla_\theta \boldsymbol{Z})^+||_2} \leq ||\omega_k \nabla_\theta L_k(t)||_2 \leq ||\nabla_\theta \boldsymbol{Z}||_2 \cdot C \quad \text{for all} \quad k = 1, 2, \dots, K, \tag{10}$$

*where* $(\nabla_\theta \boldsymbol{Z})^+$ *is the left inverse of* $\nabla_\theta \boldsymbol{Z}$*, that is,* $(\nabla_\theta \boldsymbol{Z})^+ \nabla_\theta \boldsymbol{Z} = I_d$*.*

**Proof.** First, we prove the upper bound, which is a direct consequence of the chain rule and the submultiplicative property of the 2-norm:

$$||\omega_k \nabla_\theta L_k(t)||_2 = ||\omega_k \nabla_\theta \boldsymbol{Z} \nabla_{\boldsymbol{Z}} L_k(t)||_2 \leq ||\nabla_\theta \boldsymbol{Z}||_2 \cdot C. \tag{11}$$

To obtain the lower bound we use the reasoning as before, now with the addition that, as $\text{rank}(\nabla_\theta \boldsymbol{Z}) = d$, the left inverse matrix $(\nabla_\theta \boldsymbol{Z})^+$ exists:

$$\begin{aligned} ||\omega_k \nabla_{\boldsymbol{Z}} L_k(t)||_2 &= ||\omega_k (\nabla_\theta \boldsymbol{Z})^+ \nabla_\theta \boldsymbol{Z} \nabla_{\boldsymbol{Z}} L_k(t)||_2 \\ &\leq ||(\nabla_\theta \boldsymbol{Z})^+||_2 ||\omega_k \nabla_\theta \boldsymbol{Z} \nabla_{\boldsymbol{Z}} L_k(t)||_2 \\ &= ||(\nabla_\theta \boldsymbol{Z})^+||_2 ||\omega_k \nabla_\theta L_k(t)||_2, \end{aligned}$$

which implies that

$$\frac{C}{||(\nabla_\theta \boldsymbol{Z})^+||_2} \leq ||\omega_k \nabla_\theta L_k(t)||_2. \tag{12}$$

Q.E.D.

## A.2   APPROXIMATION OF ROTOGRAD'S LOSS

In Section 4.2, we had the optimization problem

$$L_{\text{rot}}^k(R_k, \boldsymbol{d}_k, \theta) := \frac{1}{B} \sum_{i=1}^{B} ||R_k \mathcal{O}(\boldsymbol{z}_i^t) + \boldsymbol{d}_k - \mathcal{O}_k(\boldsymbol{z}_i^{k,t})||_2^2, \tag{13}$$

and claim that, by plugging in the approximation to the oracles

$$\mathcal{O}(\boldsymbol{z}_i^t) \approx \boldsymbol{z}_i^t - \beta g_i^t, \qquad \text{and} \qquad \mathcal{O}_k(\boldsymbol{z}_i^{k,t}) \approx \boldsymbol{z}_i^{k,t} - \beta g_i^{k,t}, \tag{14}$$

we could get a closed form approximation for the solution of the problem in Equation 13. Here we are going to derive this approximation. First, by plugging 14 into 13 we can split the latter into three different problems:

$$\begin{aligned} L_{\text{rot}}^k(R_k, \boldsymbol{d}_k, \theta) &= \frac{1}{B} \sum_{i=1}^{B} ||R_k \mathcal{O}(\boldsymbol{z}_i^t) + \boldsymbol{d}_k - \mathcal{O}_k(\boldsymbol{z}_i^{k,t})||_2^2 \\ &= \frac{1}{B} \sum_{i=1}^{B} ||R_k(\boldsymbol{z}_i^t - \beta g_i^t) + \boldsymbol{d}_k - (\boldsymbol{z}_i^{k,t} - \beta g_i^{k,t})||_2^2 \\ &= \frac{1}{B} \sum_{i=1}^{B} ||(R_k \boldsymbol{z}_i^t + \boldsymbol{d}_k - \boldsymbol{z}_i^{k,t}) + \beta(g_i^{k,t} - R_k g_i^t)||_2^2 \\ &= \frac{1}{B} \sum_{i=1}^{B} ||R_k \boldsymbol{z}_i^t + \boldsymbol{d}_k - \boldsymbol{z}_i^{k,t}||_2^2 \end{aligned} \tag{15}$$

$$+ \beta^2 \frac{1}{B} \sum_{i=1}^{B} ||g_i^{k,t} - R_k g_i^t||_2^2 \tag{16}$$

$$+ 2\beta \frac{1}{B} \sum_{i=1}^{B} \langle R_k \boldsymbol{z}_i^t + \boldsymbol{d}_k - \boldsymbol{z}_i^{k,t}, g_i^{k,t} - R_k g_i^t \rangle. \tag{17}$$

Note that Equation 16 only depends on the rotation matrix $R_k$ and it is a least-square problem, for which there is closed form solution. In particular, it can be shown that the solution for Equation 16 is $R_k^* = UV^\top$, where $U$ and $V$ are the two orthogonal matrices obtained in the Singular Value Decomposition of the $d \times d$ matrix obtained of multiplying the task-specific gradients $g_i^{k,t}$ with the target ones $g_i^t$, that is, $[g_1^t g_2^t \ldots g_B^t] \times [g_1^{k,t} g_2^{k,t} \ldots g_B^{k,t}]^\top = U\Sigma V^\top$. Note that in order to ensure that the matrix is a rotation, we should make sure that its determinant is one.

Similarly, it is straight-forward to show that the solution of the least-square problem for the translation $d_k$ with respect to Equation 15 is given by $d_k^* = \tilde{z}^{k,t} - R_k \tilde{z}^t$, where $\tilde{z}^{k,t}$ and $\tilde{z}^t$ are the averages over the batch of $z_i^{k,t}$ and $z_i^t$, respectively. So, instead of having in Rotograd $d_k^*$ as our translation parameter, we can have two parameters $\overline{z}^k$ and $\overline{z}$, that we update using the gradient update $\overline{z}^k - \tilde{z}^{k,t}$ and $\overline{z} - \tilde{z}^t$, respectively, and use the transformation $z_i^k = R_k(z_i - \overline{z}) + \overline{z}^k$ when doing the forward pass.

Finally, note that we can afford to ignore the error term from Equation 17 as long as our solutions in Equation 16 and 15 are good enough. The reason is that this term is nothing more but the scalar product between the residuals of the two previous problems. The error on this term will never be bigger than the maximum error committed in the other two problems.

## A.3 FURTHER EXPERIMENTAL RESULTS

In this section we present two additional plots regarding the experiments with MNIST. Specifically, Figure 4a shows the validation metrics during training for one run of Rotograd and *uniform*, showing how Rotograd outperforms the rest of the tasks in the regression metrics (being more stable in the density task), while having almost the same accuracy of the classification tasks (limited to the model capacity). More importantly, Figure 4b shows the cosine similarity between the task gradient $\nabla_z \ell_k(z)$ and the averaged gradient $\nabla_z \ell(z)$ (taking the median of the population), that is, how similar is the direction these two gradients point towards. This value is between $-1$ and $1$, being $-1$ that they point to opposite directions, $0$ that they are orthogonal (which is the most often case in high dimensional spaces), and $1$ that they are pointing to the same direction.

*Figure 4b clearly showcases the effectiveness of Rotograd in regards of homogenizing the direction of the task gradients while providing a stable training.* While keeping almost the same cosine similarity on the classification tasks, Rotograd highly increases the cosine similarity of the regression ones, being worth mentioning the case of the density task, which is pretty much orthogonal (i.e., *unrelated*) to the rest of tasks, achieving a cosine similarity of almost $0.5$ when Rotograd is applied.

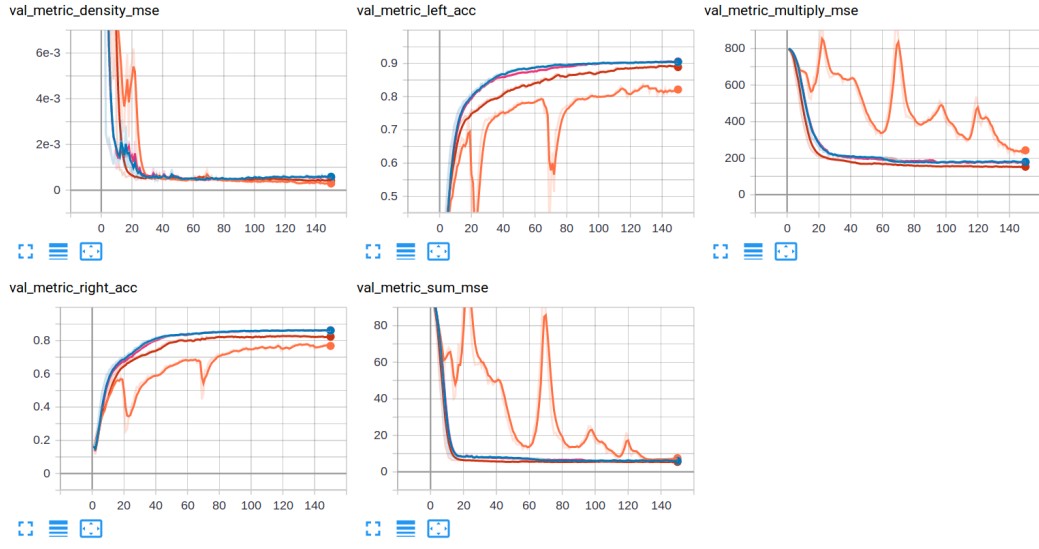

(a) Metrics over the validation set.

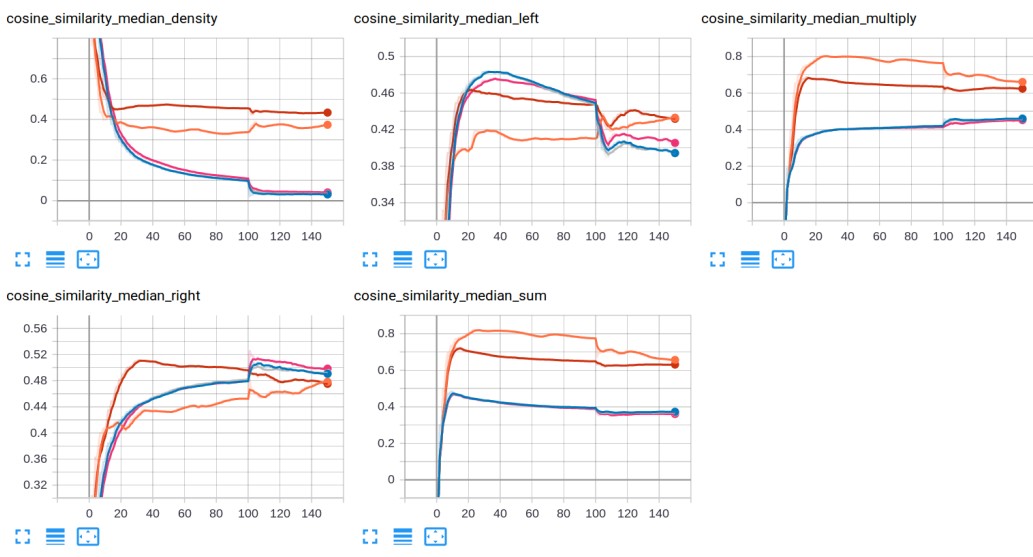

(b) Cosine similarity during training.

Figure 4: Tensorboard plots with the results of one run on the MNIST experiments during training for uniform, gradnorm $W$, gradnorm $Z$, rotograd-sgd, and rotograd. Figure (a) shows the validation metrics, where *rotograd* outperforms in the regression tasks. Figure (b) shows the (median) cosine similarity between the task gradient $\nabla_z \ell_k(z)$ and the averaged gradient $\nabla_z \ell(z)$, *clearly showing* how rotograd reduces the direction conflict among task gradients.

## A.4 CHEST

In Table 4 we show the results obtained in a similar manner as in the MNIST case. We use the same model and dataset as in Rajpurkar et al. (2017), using a single linear layer for the task-specific modules, and we train using only a 20 % of the data.

| Measure | *single task* | *uniform* | *rotograd* | *rotograd-sgd* | *gradnorm* $\mathbf{Z}$ | *gradnorm* $W$ | *d-grad* |
|---|---|---|---|---|---|---|---|
| atelectasis | $74.58 \pm 0.91$ | $77.25 \pm 0.77$ | $77.61 \pm 1.06$ | $68.98 \pm 1.87$ | $78.28 \pm 0.61$ | $77.85 \pm 0.63$ | $72.05 \pm 1.32$ |
| cardiomegaly | $62.74 \pm 5.23$ | $83.00 \pm 3.51$ | $84.43 \pm 1.85$ | $62.70 \pm 1.54$ | $85.77 \pm 0.51$ | $85.00 \pm 1.67$ | $71.47 \pm 4.20$ |
| effusion | $84.92 \pm 0.72$ | $85.57 \pm 0.29$ | $85.88 \pm 0.48$ | $77.14 \pm 1.19$ | $85.82 \pm 0.40$ | $85.56 \pm 0.64$ | $79.07 \pm 1.30$ |
| infiltration | $63.89 \pm 1.35$ | $67.65 \pm 0.39$ | $67.96 \pm 0.33$ | $63.12 \pm 0.68$ | $67.62 \pm 0.39$ | $67.31 \pm 0.49$ | $64.35 \pm 0.72$ |
| mass | $52.43 \pm 1.30$ | $76.88 \pm 1.19$ | $77.50 \pm 0.84$ | $60.89 \pm 1.51$ | $77.73 \pm 1.44$ | $78.30 \pm 1.61$ | $63.09 \pm 1.67$ |
| nodule | $53.37 \pm 2.10$ | $70.03 \pm 1.34$ | $70.50 \pm 0.70$ | $56.22 \pm 0.98$ | $70.33 \pm 1.75$ | $70.76 \pm 1.33$ | $58.70 \pm 1.01$ |
| pneumonia | $56.47 \pm 2.78$ | $70.85 \pm 1.89$ | $72.10 \pm 1.13$ | $63.11 \pm 1.75$ | $71.35 \pm 1.00$ | $71.74 \pm 1.29$ | $66.14 \pm 1.67$ |
| pneumothorax | $67.08 \pm 1.90$ | $76.18 \pm 2.33$ | $78.73 \pm 1.08$ | $64.06 \pm 2.54$ | $77.19 \pm 1.90$ | $77.52 \pm 2.55$ | $69.41 \pm 1.92$ |
| consolidation | $70.14 \pm 2.28$ | $78.28 \pm 0.53$ | $78.38 \pm 0.55$ | $74.57 \pm 0.48$ | $78.12 \pm 0.81$ | $77.01 \pm 1.27$ | $75.47 \pm 0.61$ |
| edema | $82.35 \pm 0.91$ | $86.25 \pm 0.46$ | $86.33 \pm 0.41$ | $79.16 \pm 1.17$ | $86.64 \pm 0.61$ | $86.19 \pm 0.66$ | $83.09 \pm 0.94$ |
| emphysema | $49.35 \pm 1.69$ | $78.57 \pm 2.95$ | $80.48 \pm 1.96$ | $59.61 \pm 2.18$ | $81.80 \pm 1.57$ | $82.22 \pm 1.38$ | $65.55 \pm 1.56$ |
| fibrosis | $64.69 \pm 2.06$ | $76.47 \pm 1.12$ | $77.63 \pm 1.71$ | $65.10 \pm 2.40$ | $78.07 \pm 1.36$ | $77.69 \pm 1.44$ | $69.34 \pm 0.55$ |
| pleural thick. | $55.02 \pm 1.36$ | $72.65 \pm 0.86$ | $72.34 \pm 0.97$ | $62.79 \pm 0.99$ | $71.16 \pm 2.55$ | $72.38 \pm 1.14$ | $64.59 \pm 1.06$ |
| hernia | $68.58 \pm 6.36$ | $83.50 \pm 4.29$ | $85.09 \pm 2.99$ | $62.53 \pm 7.81$ | $88.87 \pm 1.93$ | $89.23 \pm 1.84$ | $70.23 \pm 14.97$ |
| $\Delta_m$ | - | $22.15 \pm 1.37$ | $22.30 \pm 1.40$ | $3.75 \pm 1.17$ | $19.90 \pm 3.18$ | $24.02 \pm 1.28$ | $9.30 \pm 2.43$ |

Table 4: Task performance results on the ChestX-ray14 dataset for different methods. All tasks are classification tasks.

