# OpenReview forum: "Rotograd: Dynamic Gradient Homogenization for Multitask Learning"
_ICLR.cc/2021/Conference — Reject_

### Official Review · AnonReviewer4 · 2020-10-28
**Review of Rotograd**

**Rating:** 4
**Confidence:** 4

**Review:**

Summary:

This paper proposes an MTL method that encourages the gradients on shared parameters to have similar directions across different tasks. The motivation is to reduce conflicts between gradients of different tasks, so that training can proceed more smoothly, and fit multiple tasks more easily. The paper introduces a new way of thinking about this kind of method, i.e., through the lens of Stackelberg games, which could be useful in reasoning about the convergence of such methods. The method is shown to perform favorably against related methods, especially in regression settings.



Strong points:

Minimizing gradient conflict is a well-motivated way to reduce negative transfer.

The algorithm description is detailed, and should be straightforward for others to implement.

Stackelberg games are an interesting framework for thinking about methods like GradNorm and Rotograd that adaptively guide MTL training.


Weak points:

The theory is interesting at a high-level, but it is not clear that it provides insights on what makes Rotograd work. In the paper, one main takeaway from the Stackelberg games framework is that the methods converge if the leader’s learning rate is asymptotically smaller than the follower’s. This takeaway is implemented by decaying the leader’s learning rate, but it is not shown that this is a key point required for Rotograd to work. I would not be surprised if the results were unaffected if this decay were removed. If this point is really important, it should be illustrated in ablation studies. More broadly, since the point does not only apply to Rotograd, this ablation could also be done on Gradnorm and other methods. Such ablations would be one way to connect the theory to the methods.

Another main takeaway from the theory is that the rotation matrices and translation vectors should be updated with gradient descent, instead of simply replacing them each step. Intuitively, the algorithm would still make sense and be simpler if R and d were simply replaced. Experiments showing that the gradient-descent update rule is necessary would help show the value of the theory.

Similarly, the value of Proposition 4.1 is not clear. Is it to prove stability? Does this have some particular connection to Rotograd, or is it a useful fact about hard parameter-sharing methods in general?

There is one ablation “rotograd-sgd”, but it is not clear how exactly it works: Can it simply update R and d however it wants, or is Eq. 9 still used to regularize the updates in some way?

By adding the rotation matrices, it’s possible that information that would be useful to share across tasks is instead stored in these task-specific matrices. That is, conflict between tasks can beneficially lead to more general representations. Restricting R to be a rotation instead of any matrix is one step towards limiting the amount of information leakage into task-specific parameters. Is there a conceptual reason to expect that the benefits from reducing conflicts will outweigh this leakage?

The experiments are on an intentionally very small architecture, where one of the main issues is expressivity, which gives Rotograd an edge over methods that do not include an additional task-specific matrix.

In Section 5.1, does the method without Rotograd do poorly because there are no task-specific networks in that case?

Although Rotograd is motivated to reduce negative transfer, Table 1 shows that Rotograd does not reduce negative transfer, but rather improves positive transfer. That is, uniform does better than rotograd in the tasks where single-task is better than multi-task, but rotograd does better than uniform in the tasks where uniform is already better than single-task. This makes me think that the benefits of Rotograd are not coming from reducing negative transfer, but from somewhere else.

Is there an explanation for why Rotograd does not work as well for multi-class classification tasks (i.e., performs worse than all other methods for Left and Right)? Is it because the task-specific heads have larger output sizes? E.g., could it be better to have a separate rotation matrix for each class? Figure 4 in A.3 confirms that there is an issue here: the cosine similarity is not higher for rotograd for the classification tasks.

Overall, from the limited scope of the experiments it is not clear that Rotograd would provide practical advantages over competing methods. The ChestXray experiments show that although Rotograd does not hurt much, it does not help overall compared too uniform.

That said, it would be still be interesting to see whether insights from Stackelberg games could lead to practical improvements for this problem.


Minor comments:

The writing has some issues. These issues don’t make the work unclear, but they are a bit distracting. Some example suggestions for fixing distracting word choice: “palliate” -> “alleviate”, “spoiled” -> “noted”, “we have not being able to propose Rotograd, but also to derive” -> “we have proposed Rotograd, and derived”. There is also frequent non-standard mixing of em dashes with spaces and commas.

“$[r_k(t)]^\alpha$ is a hyperparameter” -> “$\alpha$ is a hyperparameter” The hyperparameter is \alpha, correct?

----

Update:

I am very happy to see the new experiments that validate the implications of the Stackelberg games theory. The main drawback of the paper is that it is not clear that direction homogenization could lead to practical improvements for multi-task learning. The additional experiments in Table 2 are useful, and suggest that much of the benefit comes from the greater expressivity due to task-specific matrices.

---

> ### Author Response · Authors · 2020-11-17
> **Response to Reviewer 4**
>
>
> We thank the reviewer for a detailed review that will help us to significantly improve the paper presentation. We will carefully revisit the manuscript to correct existing typos and improve the paper readability. We have posted a general response where we address points raised by several reviewers, including the ablation study, the "leakage" question, and a clarification regarding rotograd-sgd. Please refer to such a common response for details on these points. In addition, we below provide specific answers to the rest of your questions.
>
> ---
> **Proposition 4.1**
> We recall that the original motivation of GradNorm is to equalize the magnitude contributions of the individual tasks to the gradients w.r.t. all the shared parameters $\Theta$. However, for computational efficiency, the authors restrict the GradNorm solution to only  a subset of the parameters (corresponding in their experiments to one layer in the shared NN). In contrast, when working on the shared representation Z, we can derive a bound on the norm of the gradients w.r.t all the shared $\Theta$ for the individual tasks (refer to Proposition 4.1), and thus, as desired, apply GradNorm to the overall network by working on Z. Of course, by working on Z, we cannot make exactly equal all the gradient magnitudes but instead force them to lay in a target interval. We will clarify this point in the final manuscript.
>
> ---
> **Size of the architectures**
> Negative transfer may occur especially in mid- and small-size architectures, as the individual tasks are "forced" to cooperate (but also to compete for shared resources). Thus, in our experiments, we consider reduced architectures (still with comparable accuracy compared to the original architecture) to avoid scenarios where, due to the high number of parameters, the backbone can fit all tasks without requiring positive transfer. We would like to emphasize that, as the size of Z increases, the more likely that the gradients across tasks become orthogonal, i.e., that different tasks use disjoint subsets of the shared intermediate representation Z.
>
> ---
> **Rotograd on classification tasks**
> We agree with the reviewer that in MNIST rotograd performs slightly worse for the classifications tasks, although significantly better for the other tasks, than the uniform approach. However, such a difference in the classification tasks decreases when a more thorough  hyperparameter optimization for all the methods is performed, as shown in the following table (which contains a summary of the new results that will be replacing Table 1 in the paper):
>
> | Method          | Left 🡑 | Right 🡑 | Sum 🡓 | Multiply 🡓 | Density 🡓 |Δ 🡑 |
> | :-------------- | :-----------: | :-----------: | :---------: | :-----------: | :---------: | :----------: |
> | single task     | **93.50 (00.47)** | **90.65 (00.46)** | 6.44 (4.63) | 159.08 (6.16) | 1.62 (1.78) |              |
> | uniform         | 90.15 (00.53) | 86.65 (00.41) | 5.14 (0.33) | 149.21 (5.97) | 0.51 (0.02) | 0.06 (0.11)  |
> | rotograd        | 89.01 (00.87) | 84.62 (01.19) | **4.54 (0.19)** | **134.95 (5.92)** | **0.23 (0.04)** | **0.18 (0.06)**  |
>
> The above results correspond to a learning rate of 0.02 for the leader and an exponential decay of 0.99 per iteration.
>
> We believe that the slight deterioration in classification accuracy is due to the limited capacity of the NN, which trades-off the performance across all tasks. This can be explained by per-task learning dynamics shown in Fig. 4(a) of Appendix A3, where we see that uniform aggressively optimizes both classification tasks, the other tasks are learned at a lower pace. A similar behavior is observed when looking at the cosine similarities for the individual tasks  in Fig. 4(b), where we can observe that while the gradient of the classification tasks is well aligned with the overall gradient evaluation (cosine similarity approx. 0.5), this is not the case for the rest of tasks (being, e.g., the cosine similarity of the density below 0.1). In contrast, rotograd forces a similar cosine similarity for all tasks, and thus  eases that all tasks are learned at a similar pace (although more slowly for the classification compared to uniform). In order words,  rotograd makes the cosine similarity comparable across all tasks (which means worsening the classification performance), whereas in uniform the density task is completely orthogonal to the others.

---

### Official Review · AnonReviewer1 · 2020-10-29
**The writing needs improvement. The proposed idea is not well justified. The empirical results are weak.**

**Rating:** 4
**Confidence:** 4

**Review:**

This paper presents an extension of Gradnorm to address task conflicting due to discordant gradient direction. Specially it introduces a rotation matrix to rotate the hidden representation from the last shared layer. The authors put the proposed method in the context of game theory to show stability and convergence of the training, which might be of merit.

The writing of the paper doesn’t meet the publication standard, needing major work to improve. There are many typos and awkward sentences, hindering understanding of their work. Also, there are many places that need clarification, for example, in Proposition 4.1, the inverse of the gradient of Z with respective to \theta needs to be calculated. So, what is the shape of this gradient matrix? How it is necessarily to be a square matrix? What ||\Delta_{\theta} Z|| represents? the F-norm? There is lack of adequate explanation of the motivation behind the objective in Eq. (6). By reading the paper, I have no idea about the two oracle functions, and why they are defined in the way shown in Eq. (8).

Eq. (3) is inaccurate, not aligning with that proposed in the GradNorm paper for the computation of L_{grad}^k.

Eq. (9) is problematic. Why R_k z_i^t does not appear in the objective function of the first optimization problem? If this is because z_i^{k,t} = R_k z_i^t + d_k, then the objective in the second optimization problem would be just 0.

Why operating on z instead of the gradient in Gradnorm can resolve the discordant gradient issue among tasks is not properly justified.

The reported empirical results are weak and do not support this method works as claimed.

---

> ### Author Response · Authors · 2020-11-17
> **Response to Reviewer 1**
>
> **Clarity of the paper**
> As suggested by the reviewer, we will carefully review the writing of the paper to improve its readability and to make it more self-contained by providing all the necessary details on both the motivation and technical details of our approach.
>
> ---
> **Proposition 4.1 and the role of Z**
> The point of Prop. 4.1 is that we can bound the norm of the gradient w.r.t. the shared parameters (the original goal of GradNorm) by equalizing the norm of the gradient w.r.t the shared representation Z. Such bound is given in Eq. 5, which depends on  the inverse of the gradient matrix. The gradient matrix is of the size of the number of parameters  times the size of the shared representation Z, and thus in general case is not invertible (as it is not squared). Fortunately, our theoretical results still hold when the gradient matrix  is left-invertible, which does not require a squared matrix as it only requires that the rank for the gradient coincides with the dimensionality of Z (which is in general significantly smaller than the number of parameters). Moreover, we point out that we do not need to restrict ourselves to a particular norm, as all norms are equivalent in finite-dimensional spaces and thus do not change the validity of our results. We will clarify this in the revised version of the paper.
>
> ---
> **Oracle formulation (Eq. 8) and Leader objective (Eq. 9)**
> There are two types of oracle functions in our formulation, which predict the next evaluation point of respectively the shared and the individual tasks' representation and are approximated in Eq. 8. These approximations correspond to a step of gradient descent (as detailed before Eq. 8), and are necessary in order to be able to solve the leader objective, i.e., to find the optimal  transformation of the shared representation into the individual tasks' representation in the next iteration of the learning algorithm. We will clarify this in the revised version of the paper.
>
> When considering the oracle approximation in Eq. 8, the leader objective in Eq. 6 readily split into the two objectives in Eq. 9, plus a residual term that tends to zero when the two individual objectives in Eq. 9 are solved with zero error (refer to Eq. 18 in appendix A2 for the exact relationship). When Eq. 9 cannot be perfectly solved, the solution of Eq. 9 approximates (up to a mismatch) the solution of Eq. 6. We refer the reviewer to appendix A2 for further details.
>
> ---
> **GradNorm formulation**
> We do not see any difference between Eq. 3 in our paper and Eq. 2 in the GradNorm paper*, except for the adaptation to our notation and for the fact that in Eq. 3 we only consider one particular task (i.e., we have removed the summatory over tasks in Eq. 2 of GradNorm).
>
> *https://arxiv.org/pdf/1711.02257.pdf

---

### Official Review · AnonReviewer3 · 2020-10-29
**Interesting idea but weak experiment implementation and lack of motivation for the proposed method**

**Rating:** 4
**Confidence:** 4

**Review:**

In the paper, Rotograd is proposed as a new gradient-based approach for training multi-task deep neural networks based on GradNorm. GradNorm is first formulated as a Stackelberg game, where the leader aims at normalizing the gradient of different tasks and the follower aims at optimizing the collective weighted loss objective. Under this formulation, one can utilize theoretical guarantees of the Stackelberg game by making the leader have a learning rate that decays to zero faster than the follower. To further account for the different gradient directions, a learnable rotation and translation are applied to the representation of each task, such that the transformed representation match that of the single-task learning. By adding an additional term accounting for learning this rotation, the leader in the Stackelberg game will minimize the loss to homogenize both the gradient magnitude and match the representation to single-task learning as close as possible.

In general, I find the direction of gradient homogenization for multi-task learning very important and interesting. The paper provides an interesting perspective through the Stackelberg game formulation, which provides a framework for selecting the learning rate of GradNorm type of gradient homogenization methods. The other contribution of the paper is a learnable task-specific rotation that aligns the task gradients with single-task learning. The proposing of a learnable rotation matrix seems an interesting idea, although I am not sure if it has been proposed previously for multi-task learning.

I find the first contribution of formulating the problem as a Stackelberg game to be interesting and novel. However, in terms of the second contribution, I have some concerns about whether it makes the most sense by aligning the transformed representation with that of single-task learning. For MTL, one of the key benefits is learning a better representation by sharing it across different tasks to encourage helpful transfer between the tasks; by constraining the transformed representation to be as close to the single-task learning representation, it might limit the transfer between tasks since the representation are constrained to be equivalent to that learned by single-task learning. I think it is helpful to think about using rotation invariant representations for aligning the gradient directions, but it is questionable to align it to that of the single-task learning.

Another major concern is about the experimental results, full experiments are only conducted on one real-world dataset. The experiment on the second dataset seems to be very preliminary, which might not be sufficient to justify the proposed method empirically. Also on the second dataset, it seems the two different implementations of Rotograd have a large discrepancy in the results, which might need more investigation about why this happens. Meanwhile, many ablation studies seem to be missing. I am mostly interested to see experiments that validate the Stackelberg game formulation, for example by using different learning rates for the leader and the follower. Also, it would be interesting to see how the proposed Rotograd compares with pure GradNorm on gradient direction alignment. Overall, I feel the experiments are not complete for validating the effectiveness of the method.

Some minor points: the description of d-grad method seems to be missing. Also, Yu et. al [2020] also deals with gradient aligning for MTL which could be considered as a baseline to compare with.

Yu, T., Kumar, S., Gupta, A., Levine, S., Hausman, K., & Finn, C. (2020). Gradient surgery for multi-task learning. arXiv preprint arXiv:2001.06782.

--------After author's response----------

I am not fully convinced by the explanation of the motivation behind rotation matrix, in particular why it is aligning with the single-task learning, which is counter-intuitive. The authors provided more ablation studies, however, the evaluation on datasets is still quite preliminary with some questions remaining (such as why there is a discrepancy between the two versions of Rotograd on the second dataset). Therefore I am keeping my original score.

---

> ### Author Response · Authors · 2020-11-17
> **Response to Reviewer 3**
>
> First of all, we would like to thank the reviewer for such an accurate summary of our work. We are delighted to observe interest in this novel formulation based on Stackelberg games, which we believe can shed some light on the dynamics of MTL methods such as GradNorm and Rotograd.
>
> ---
> **2nd contribution - Rotograd**
> Up to the best of our knowledge, the closest approach to ours is "d-grad via architecture", which implements using a NN-architecture an affine transformation of the output of every hidden layer in the multitask network with the aim of avoiding conflicting gradients. However, d-grad suffers from two major limitations in comparison to rotograd. First, in its formulation there is not an explicit objective on the gradient alignment, and thus, it does not provide any theoretical guarantees. As a consequence, it is not clear if the better performance shown in the empirical evaluation is due to avoiding conflicting gradients or to the additional expressiveness of the model (as there is an additional affine transformation per layer). Second, d-grad does not impose restrictions on the affine transformation NNs (or equivalently, on the individual gradient magnitudes), which may still result in a negative transfer between tasks due to the disparities in the individual gradient magnitudes (problem addressed by Gradnorm and our extension, i.e., rotograd). We will provide a detailed description of d-grad and its (dis-)similarities with rotograd in the revised version of the manuscript
>
> ---
> **Experimental evaluation and representation aligment**
> Please refer you to our general response for a detailed response on the empirical evaluation of rotograd, which includes an ablation study to validate the Stackelberg formulation, and to the question regarding the representation alignment.

---

### Author Response · Authors · 2020-11-17
**General response to all reviewers (1/2)**

We would like to thank the reviewers for providing us with such helpful feedback, which will ultimately improve our work. In general, all reviewers agreed on the importance of the problem we address, for example, reviewer 4 said that “minimizing gradient conflict is a well-motivated way to reduce negative transfer”. Moreover, all three reviewers expressed their interest and acknowledged the merit of the novel perspective based on Stackelberg games proposed in this work. Since several questions and concerns seem shared among the reviewers, we next address the common concerns. Specific answers to individual reviewer's comments are provided in separate answers.

---

**Representation alignment / Rotation matrix interpretation**
We would like to remark that the goal of rotograd is to find both a common representation shared across all tasks, and a way to transform it to the individual tasks' representations. Our interpretation is that the rotation matrix acts as a “translator” between the common representation (which aims to jointly capture all the tasks and their relationships) and the task-specific ones (whose only objective is to optimize its individual loss function). While the shared representation performs the same role as any other shared representation in MTL, i.e., easing positive transfer ("leakage") between tasks, the rotation that maps it to the individual tasks' representations avoids potential negative transfer among tasks due to the "disagreements" among the individual objectives.

Importantly, we would like to add that while we agree with the reviewer that, in absence of the GradNorm loss in Eq. 7, a perfect solution of Eq. 6 would decouple the learning between tasks preventing positive transfer among tasks, such a perfect solution can only be found in trivial settings, where the rotation transforms the shared representation into the task-specific one for all the observed samples (not only the samples in the current batch) with zero error. This is highly unlikely (if even possible) in our settings due to the stochasticity of the inputs (an in turn in the per-observation gradient evaluation),  the stochasticity introduced by the batch, and the imperfect oracle functions (i.e., approximation of the oracles in Eq. 8). Moreover, as we also perform iterative (SGD) updates in the leader, even if a zero error solution of Eq. 6 existed, by the time that the the follower reaches its optimum, information across tasks would have already been shared in the shared representation (and thus, the shared parameters).

---
**Clarifications / Rotograd-sgd**
In order to improve the accessibility and readability of the paper,  we will add more extensive explanations and further clarifications in the revised version of the paper. The description rotograd-sgd deserves  special attention, as it seems to not be stated clearly enough in our experiments. The baseline rotograd-sgd shares the same learnable parameters as the standard rotograd and optimizes the same two objectives in Eq. 6. However, instead of solving Eq. 6 in closed-form (as the proposed rotograd does) and setting the gradient as the difference between the current point and this solution, it instead relies  on automatic differentiation to evaluate the gradient of the parameters.

---

> ### Author Response · Authors · 2020-11-17
> **General response to all reviewers (2/2)**
>
> **Ablation study**
> We agree with the reviewers that the paper would benefit from an ablation study to better understand the role of each element in the problem formulation. Thus, we will include in the revised version of the paper several new sets of experiments:
> First, to better understand the implications of the Stackelberg results on the stability of the training, we perform experiments with different learning rates and decays for the leader. As a special case we consider SGD with learning rate 1 and no decay, which corresponds to the case where we directly set the parameters at each step to closed-form solutions of Eq. 9. The following table provides a subset of the results we plan to add in the paper (for MNIST  with different seeds, initial learning rates equal to 0.02 and 0.001 for respectively the leader and follower, and with different decays):
>
> | decay | Left 🡑 | Right 🡑 | Sum 🡓 | Multiply 🡓 | Density 🡓 | Δ 🡑   |
> | :-------------- | :--: | :---: | :--: | :------: | :-----: | :----------: |
> | 0.9 | **89.46 (0.57)** | **85.90 (0.65)** | 4.64 (0.25) | **133.97 (7.20)** | 0.26 (0.06) | **0.18 (0.06)** |
> | 0.99 | 88.99 (0.81) | 84.92 (0.97) | **4.58 (0.16)** | 135.19 (4.35) | **0.24 (0.05)** | **0.18 (0.06)** |
> | 0.999 | 85.51 (1.55) | 80.09 (1.86) | 4.79 (0.18) | 142.07 (3.86) | 0.24 (0.08) | 0.15 (0.06) |
> | 0.9999 | 84.56 (1.65) | 79.46 (1.80) | 4.81 (0.23) | 142.64 (4.07) | 0.24 (0.06) | 0.15 (0.06) |
> | 1.0 | 83.55 (1.69) | 79.31 (2.43) | 4.88 (0.20) | 143.41 (3.17) | 0.25 (0.13) | 0.13 (0.09) |
>
> Here, we can observe that the faster the leader learning rate decays, the better the MTL results. Similarly, results where we directly update Rotograd’s parameters instead of performing iterative updates show this instability problems, as shown in the following table:
>
> | Method          | Left 🡑 | Right 🡑 | Sum 🡓 | Multiply 🡓 | Density 🡓 | Δ 🡑 |
> | :-------------- | :-----------: | :-----------: | :---------: | :-----------: | :---------: | :----------: |
> | single task     | **93.50 (00.47)** | **90.65 (00.46)** | 6.44 (4.63) | 159.08 (6.16) | 1.62 (1.78) |              |
> | uniform         | 90.15 (00.53) | 86.65 (00.41) | 5.14 (0.33) | 149.21 (5.97) | 0.51 (0.02) | 0.06 (0.11)  |
> | rotograd        | 89.01 (00.87) | 84.62 (01.19) | **4.54 (0.19)** | **134.95 (5.92)** | **0.23 (0.04)** | **0.18 (0.06)**  |
> | rotograd (lr=1) | 64.73 (4.83)  | 58.55 (5.94)  | 6.11 (0.26) | 193.93 (9.53) | 0.89 (0.21) | -0.23 (0.21) |
>
> In conclusion,  the above results confirm the need of a slow-learning leader (see the section of necessity vs sufficiency below for an intuitive explanation). These are illustrative results, and more detailed versions of them will be added to the paper.
>
> In addition, we plan to include additional experiments to:
> Understand the sensitivity of the model capacity to negative transfer, by testing identical architecture but with different number of parameters (and thus, different model capacities).
> Show the potential of rotograd. By performing a more thorough hyperparameter search on the learning rate and decay of the leader, we can improve the results shown in Table 2 of the original paper.
> Better understand how existing methods affect the cosine similarity of the gradients. To this end, we will include in Fig. 4 of the appendix the rest of the comparing methods.
>
> By performing these changes, we expect to cover all the experimental concerns and further encourage the adoption of rotograd by the community.
>
> ---
> **Sufficiency vs. necessity**
> We find it necessary to remark that the stability results obtained from the Stackelberg formulation provide sufficient (and not necessary) conditions to achieve this stability. While we are able to consistently reach stable solutions in the toy examples (Fig. 3) by directly updating rotograd’s parameter to their closed-form solution, making sure that the leader is the slow learner (thus ensuring stability) becomes more and more important as we move to more complex datasets such as MNIST and, specially, ChestXRay. As mentioned above, we will include stability experiments to show the importance of the leader’s learning speed in non-trivial scenarios where this aspect becomes relevant.

---

> > ### Author Response · Authors · 2020-11-25
> > **New revision of the paper**
> >
> > We would like to let the reviewers know that a new version of the paper has been uploaded, implementing the main changes and comments requested by the reviewers. The main changes of the new manuscript are the following:
> > - Typos and wrong use of em dashes have been correct and, in general, the text has been polished to improve readability.
> > - Proposition 4.1 has been replaced by a paragraph explaining what are the implications that this proposition have in supporting the application of GradNorm over the shared representation. Prop 4.1 is now in the appendix and has been further improved, only requiring the gradient matrix to have a left-inverse instead of being invertible.
> > -  Additional comments have been added regarding how to simulate the oracles and how to obtain Rotograd's approximated objectives from its objective function.
> > - Experiments now include three ablation studies:
> >    - First, we study the effect that the leader's learning speed has on the training of the MNIST experiments, showing that a slower learner benefits the overall results (Table 1).
> >   - Second, the special case where the parameters of Rotograd are updated directly using the closed-form solutions is considered. Showing (Table 1) that the training becomes highly unstable, obtaining poor results. This further support the use of iterative updates.
> >   - Third, using the same experimental setup we study how the model capacity influences negative transfer. Specifically, we show (Table 2) that the effect of negative transfer becomes more noticeable as we restrict the model capacity, since task cooperation is then required to further improve their results.
> > - A more extensive description of the considered methods (e.g. rotograd-sgd) is given, making clearer the differences across methods.
> > - Results regarding the previous experiments on MNIST and Chest have been updated according to a better tuning of the leader's learning speed.
> > - Figure 4 of the appendix have been updated, now including all the different methods that were used in the main paper, rather than having only uniform and Rotograd.
> >
> > We would like to thanks once again the reviewers for their useful feedback and we hope these changes are well received.

---

### Decision · Program_Chairs · 2021-01-07
**Final Decision**

**Decision:**

Reject

**Comment:**

The paper is proposing a novel representation of the GradNorm. GradNorm is presented as a Stackelberg game and its theory is used to understand and improve the convergence of the GradNorm. Moreover, in addition to the magnitude normalization, a direction normalization objective is added to the leader and a rotation matrix and a translation is used for this alignment. The paper is reviewed by three knowledgable reviewers and they unanimously agree on the rejection. Here are the major issues raised by the reviewers and the are chair:
- The motivation behind the rotation matrix layers is not clear. It should be motivated in more detail and explained better with additional illustrations and analyses.
- Empirical study is weak. More state of the art approaches from MTL should be included and more realistic datasets should be included.
- The proposed method is not properly explained with respect to existing methods. There are MTL methods beyond GradNorm like PCGrad and MGDA (MTL as MOO). These methods also fix directions. Hence, it is not clear what is the relationship of the proposed method with these ones.

I strongly recommend authors to improve their paper by fixing these major issues and submit to the next venue.